# Current Assessment of Water Quality and Biota Characteristics of the Pelagic Ecosystem of the Atlantic Sector of Antarctica: The Multidisciplinary Studies by the Institute of Biology of the Southern Seas

**Natalia Mirzoeva \*, Tatiana Polyakova, Ernest Samyshev, Tatiana Churilova, Vladimir Mukhanov, Alexandr Melnik**  **, Vladislav Proskurnin, Evgeny Sakhon, Elena Skorokhod, Olga Chuzhikova-Proskurnina, Elena Chudinovskih, Natalia Minkina, Natalia Moiseeva, Victor Melnikov, Artem Paraskiv, Lidia Melnik and Tatiana Efimova**

A. O. Kovalevsky Institute of Biology of the Southern Seas (IBSS), Russian Academy of Sciences, 299011 Sevastopol, Russia
\* Correspondence: natmirz@mail.ru; Tel.: +7-(978)739-80-79

**Abstract:** Comprehensive studies of the ecosystem of the Atlantic sector of the Antarctic were carried out in the period between 2020 and 2022, during the 79th and 87th sea expeditions on the R/V "Akademik Mstislav Keldysh". Concentrations of soluble forms of 15 trace elements, except Mo, in surface waters of the Southern Ocean were matched the lower limit of their background concentrations in oceanic waters. A high spatial variability of chlorophyll *a*—an indicator of phytoplankton biomass, which is the main food object of the Antarctic krill—was revealed. In the Bransfield Strait, the abundance of bacterioplankton exhibited a weak relationship with water temperature and a pronounced dependence on chlorophyll fluorescence. It was determined, by using the bioluminescence method, that the largest concentrations of larvae and juveniles of krill were noted in the Bransfield Strait, on the shelf of the Antarctic Peninsula. Against the background of a decline in krill abundance in recent years, there has been an intensive development of *Salpa thompsoni*, the main food competitor of krill. New data on the distribution of mesoparasitic copepods on endemic fish of the Southern Ocean were also obtained, and their pathogenic effect on the body of fish hosts has been revealed.

**Keywords:** *Euphausia superba*; Salpa thompsoni; mesoparasitic copepods; bacterioplankton; bioluminescence; chlorophyll *a*; optically active components; trace elements; heavy metals; Atlantic sector of the Antarctic



## 1. Introduction

The Southern Ocean is one of the most productive and ecologically clean areas of the World Ocean. This area, with large-scale inventory of marine biological resources, is a fishing area (under international fishing agreements). Antarctic waters are differentiated by high biological productivity with relatively low biodiversity [1–3]. Whales, penguins, seals, seabirds, fish and numerous invertebrates coexist in the Southern Ocean ecosystem. These organisms survive predominantly by feeding on Antarctic krill (*Euphausia superba* Dana, 1852). Antarctic krill is an essential target of Antarctic fisheries, and the basis of the diet of its many consumers. Krill stocks in the Southern Ocean were, until recently, estimated at hundreds of millions of tons [2,4–8]. Now, assessment of the status of krill communities is one of the priority research areas in the Antarctic sector of the Atlantic Ocean.

According to international agreements, access to Antarctic resources is granted to those countries that carry out scientific research in this unique region of the planet [9,10]. The expediency of conducting research on bioresources in the Antarctic is due to the prospects for their development and the need to create a theoretical basis for fishing, primarily for Antarctic krill and several species of fish [2,3,11,12], as well as the impacts of climate

change and anthropogenic action in the contemporary period [13–15]. At the same time, the methodological basis of biological research should be a multidisciplinary monitoring of the Antarctic ecosystem [16].

The 79th and 87th cruises of the research vessel (R/V) "Akademik Mstislav Keldysh" to the Atlantic sector of the Antarctic took place from 30 November 2019 to 8 May 2020, and from 7 December 2021 to 6 April 2022, respectively [17,18]. The organization and management of these expeditions was carried out by the P.P. Shirshov Institute of Oceanology of the Russian Academy of Sciences (RAS), whose scientific fleet includes the R/V "Akademik Mstislav Keldysh". These marine expeditions to study the natural complexes of the Antarctic waters were carried out within the framework of the international obligations of the Russian Federation as a party to the Antarctic Treaty [10], as well as the Convention on the Conservation of Antarctic Marine Living Resources [9]. During the cruise, some fundamental tasks, assigned to the scientists of Russia in a number of strategic documents, were addressed [19,20].

Research workers of six Departments of the A. O. Kovalevsky Institute of Biology of the Southern Seas (IBSS) RAS (Department of Radiation and Chemical Biology; Department of Marine Ecosystem Functioning; Department of Ecological Parasitology; Department of Plankton; Department of Biophysical Ecology; Scientific Research Center (SRC) "Geomatics") participated in study of the Atlantic sector of the Antarctic during the 79th and 87th cruises of the R/V Akademik Mstislav Keldysh. All specific scientific research activities were combined into multidisciplinary studies carried out at the level of the IBSS teams, and the research topics of all scientific organizations of Russia participating in these marine cruises were combined.

The aim of the scientific research of the IBSS team in 2020 and 2022 was to obtain new data for a current assessment of the ecological state of the Antarctic ecosystem in relation to the content of trace elements in sea water, including heavy metals. This was studied in order to determine the spatial, structural and functional characteristics of bacterio- and phytoplankton as the main food resources for Antarctic krill, as well as to understand the spatial and quantitative variability of zooplankton, primarily Antarctic krill. Additionally, the objective of parasitological research was to fill in significant gaps in the study of the parasite fauna of Antarctic animals.

To achieve the aim of this investigation, the following tasks were addressed: (1) determination and analysis of the concentrations of soluble forms of a number of trace elements (including heavy metals) in the surface waters of the Atlantic sector of the Antarctic in order to assess the current environmental situation in this area, as well as to identify possible sources of the input of the studied elements into the considered water area; (2) quantifying the total abundance of bacterioplankton and high nucleic acid (HNA) bacteria, and analyzing their relationship with chlorophyll fluorescence in the central part of the Bransfield Strait; (3) study of the variability of the spectral absorption of light by all optically active components and photosynthetic characteristics of phytoplankton; (4) observation and analysis of bioluminescence of Antarctic krill and salp (*Salpa thompsoni* Foxton, 1961) as important elements of the functioning of the pelagic community; (5) conducting studies of contemporary biological productivity, structure and spatial organization of Antarctic krill; (6) studying the pathogenic influence and distribution of mesoparasitic copepods in endemic bathypelagic fishes of the Southern Ocean.

Antarctic krill and Antarctic fish are valuable fishery resources for all mankind [9,10]. Therefore, this research has unconditional scientific, ecological and practical bioresource significance.

## 2. Materials and Methods

### 2.1. Trace Elements

The water samples for trace elements analysis were taken during the 87th cruise of the R/V "Akademik Mstislav Keldysh" ("AMK"), organized by the P.P. Shirshov Institute of

Oceanology of RAS. The Antarctic portion of the expedition was held in January–February 2022 (Figure 1).

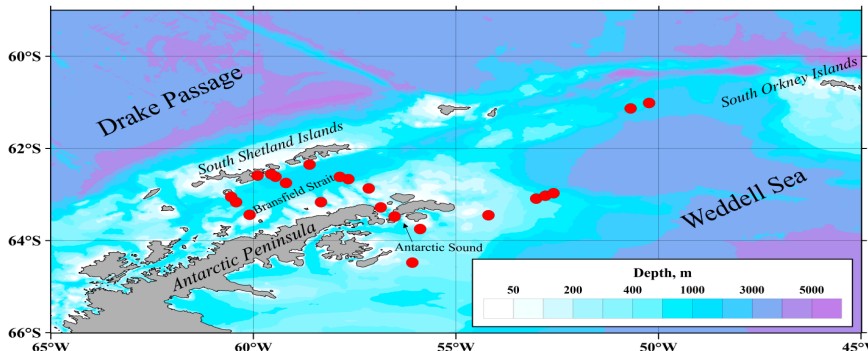

**Figure 1.** Location of water sampling sites for trace elements analysis in 87th cruise of R/V "Akademik Mstislav Keldysh".

The cruise undertook 3 transects across the Bransfield Strait in its eastern, central and western parts, a transect from Bransfield Strait through the Antarctic Sound to the NW part of the Weddell Sea (where the anomalous phytoplankton bloom observed at the time of sampling [21]), and a transect through the NW Weddell Sea in the NE direction. The results of this study were compared to those obtained in 2020 [22,23].

Surface water was sampled by an acid-cleaned plastic sampler directly from the shipboard on a rope. Seawater was vacuum-filtered through a 0.45 μm nitrate cellulose membrane immediately after sampling and acidified with high pure nitric acid to pH < 2. Acidified samples were stored in a refrigerator at 2–3 °C until they were treated in the ship's laboratory. The onboard sample treatment technique which was applied resulted in the extraction of selected trace elements with carbon tetrachloride (CTC or $CCl_4$) in the form of diethildithiocarbamate (DDC) complexes in 3 replicates for each sample, followed by re-extraction of elements by destruction of the complexes with concentrated nitric acid [23,24]. The set of measured elements was determined by their ability to forma complex with a DDC ion extractable with CTC, and included 15 elements: Be (beryllium), V (vanadium), Fe (iron), Co (cobalt), Ni (nickel), Cu (copper), Zn (zinc), As (arsenic), Se (selenium), Mo (molibdenium), Ag (silver), Cd (cadmium), Sb (antimony), Tl (thallium), and Pb (lead). Concentrated solutions of each sample were stored in a refrigerator during the cruise, and were then transported to the IBSS Center for Collective Use "Spectrometry and Chromatography", where concentrations of the elements were measured by the ICP-MS technique on a PlasmaQuant MS Elite (Analytik Jena AG) mass-spectrometer [25,26]. The spectrometer was calibrated using a standard solution: "Multi-element calibration standard IV-28, $HNO_3$/HF, 125 mL" (Inorganic Ventures). The mode of measurement with the mass-spectrometer included 7 replicates of 10 scans for each identified element from 10,000 to 100,000 μs, depending on its expected concentration. The calculation and registration of the measurement results were carried out in accordance with GOST R 56219-2014 and RD 52.10.243-92 [24,25]. The average relative determination error was not higher than ±10%.

## 2.2. Bacterioplankton

The data on bacterioplankton and chlorophyll *a* fluorescence were collected on January 21, 2020, over a southward transect across the central Bransfield Strait at 7 stations (st. 6587, 6590, 6591, 6592, 6593, 6594, and 6595) (Figure 2).

The length of the transect from Greenwich Island (South Shetland Islands) to the shelf of the Antarctic Peninsula was 93 km. Water samples were taken at various points, from 5–7 horizons from the surface to 190 m of depth, depending on the hydrological structure and fluorescence distribution. Bacterial abundances in the samples were estimated by flow cytometry. Counts were performed using a Beckman Coulter flow cytometer (Cytomics

FC 500, Beckman Coulter Inc., Brea, CA, USA) equipped with an air-cooled blue laser
(15 mW, 488 nm) and a standard filter setup. Aliquots of 1mL water samples previously
fixed with formaldehyde (2% final conc.) were stained with SYBR Green I (Molecular
Probes Inc., Eugene, Oregon, USA), following the procedures described by [27]. SYBR-
Green I fluorescence, at the green channel FL1 (525 nm), was considered proportional to
intracellular nucleic acid content, and was interpreted as a measure of bacterial cell-specific
metabolic activity [28]. Consequently, high nucleic acid (HNA) and low nucleic acid (LNA)
bacteria were gated on the cytograms (Figure 3).

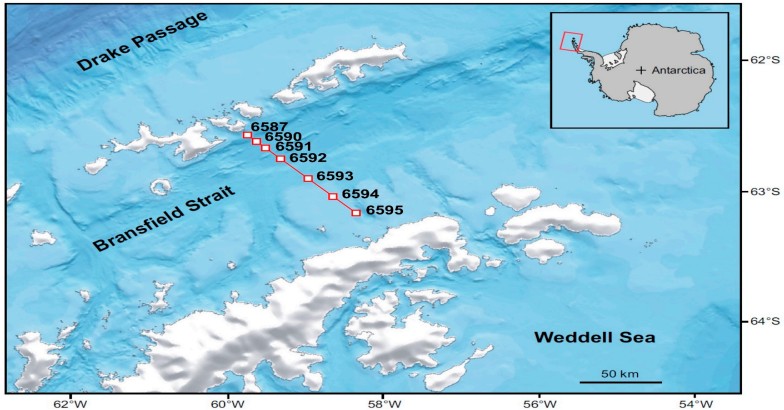

**Figure 2.** Locations of oceanographic stations in the Bransfield Strait (79th cruise of R/V "Akademik
Mstislav Keldysh", January 2020) where bacterioplankton samples were collected at 5 to 7 depths in
austral mid-summer.

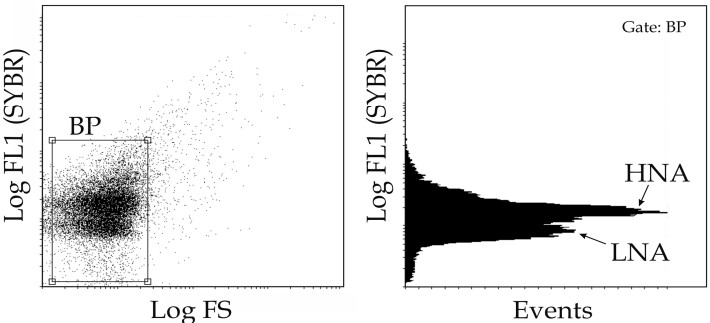

**Figure 3.** Gating total bacterioplankton (BP), HNA-, and LNA-bacteria in the space of direct light
scattering (FS, cell size), and green fluorescence (FL1, SYBR).

Temperature, conductivity, and pressure data were collected using an SBE911 CTD
attached to a Carrousel system with 24 5-L Niskin bottles for water sampling. Profile
measurements of chlorophyll *a* fluorescence were conducted using a PUM-200 transparency
meter equipped with a Minitracka-II fluorimeter (Chelsey Instruments, Molesey, UK).

### 2.3. Spectral Bio-Optical Properties and Productive Characteristics of Phytoplankton

Water samples were taken using 5-L bottles from different depths, which were chosen
based on chlorophyll a fluorescence and temperature profiles. A map of the sampling
scheme is shown in Figure 4.

Optical densities of the samples were measured with a dual-beam spectrophotome-
ter, Lambda 35 (PerkinElmer), equipped with an integrating sphere. Chlorophyll *a* and
phaeopigment concentrations were measured by the spectrophotometric method [29,30].
The light absorption of particles ($a_p(\lambda)$), phytoplankton ($a_{ph}(\lambda)$), non-algal particles ($a_{NAP}(\lambda)$),
and colored dissolved organic matter ($a_{CDOM}(\lambda)$) were measured in accordance with NASA
protocols [31]. For $a_{NAP}(\lambda)$ and $a_{ph}(\lambda)$ determination, pigments were bleached by the
method used in [32], and β-correction was performed in accordance with [33]. The $a_{NAP}(\lambda)$

and $a_{CDOM}(\lambda)$ data were fitted by exponential function [34]. Slope coefficients ($S_{NAP}$ and $S_{CDOM)}$ were estimated for the wavelength domain of 400−700 nm and 350–500 nm, respectively. The maximum quantum yield of photosynthesis ($\phi_{max}$) was calculated [35] based on the maximum quantum yield of PSII measured with PAM fluorimeter (Water PAM-II, Walz) [36,37]. The values of the quantum yield of photosynthesis ($\phi_z$) at different depths were calculated based on $\phi_{max}$ and saturating light intensity ($I_k$) [38]. $I_k$ was determined based on light curves of the relative electron transport rate, measured with a PAM fluorimeter.

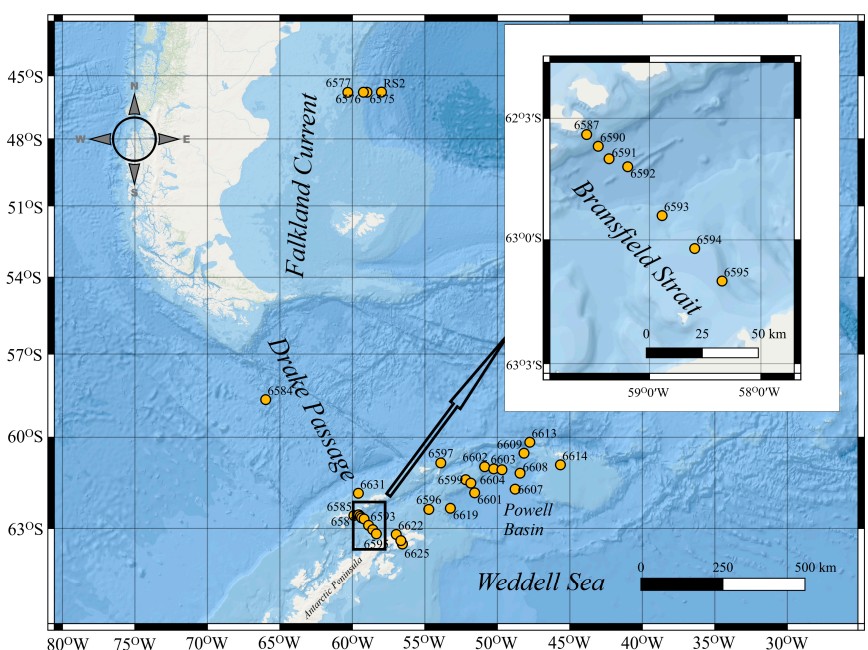

**Figure 4.** The map-scheme of sampling for scientific research of the spectral bio-optical properties and productive characteristics of phytoplankton.

## 2.4. Bioluminescence

The multiple sounding method was applied [39] using an autonomous hydrobiological system «Salpa» [40,41]. The «Salpa» was designed to study the intensity of the bioluminescence of organisms in the World Ocean's layer, at 0–250 m depths, using the method of multiple vertical sounding at a speed of up to 1.2 m·s$^{-1}$. "Salpa" allows simultaneous measurements of bioluminescent potential, temperature, hydrostatic pressure, turbidity, electrical conductivity, and photosynthetically active radiation to be taken. Among the existing methods for measuring bioluminescence signals in pelagial environments (towing photometers, hanging them on a given horizon, installing them using special trusses on the bottom, etc.) in recent years, the method of sounding the water column has been recognized as the most promising and accurate. The essence of the applied method is that hydrobionts, which make the main contribution to the formation of the bioluminescent potential of the pelagic zone, generally flash with light when mechanically stimulated. The method of collecting and processing the data using the "Salpa-M" complex has previously been described in detail [40,41]. The bathyphotometer "Salpa" creates a standard level of mechanical excitation of bioluminescence, which makes it possible to achieve an accurate comparison of the measured values of the vertical structure of the bioluminescence field (BF) when performed in different regions and under various conditions. The data on bioluminescence were collected over a transect across the central Bransfield Strait at 9 stations (Figure 5).

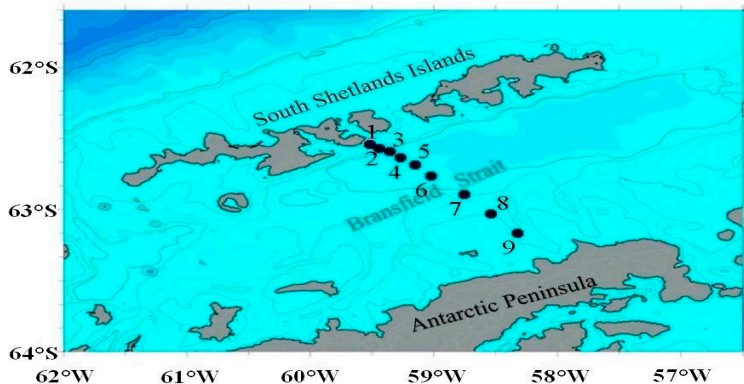

**Figure 5.** Location of bioluminescence stations in the Bransfield Strait.

### 2.5. Antarctic Plankton

The investigations on distribution krill and salps were studied during the 87th cruise of the R/V "Akademik Mstislav Keldysh" in January and February 2022 in the western zone of the Atlantic sector of Antarctica. A total of 31 tows were performed: 21 by the DSN plankton net and 10 by the Isaacs-Kidd trawl in the Samyshev–Aseev modification [42,43]. A sample of at least 300 individuals was selected for biological analysis. If there were fewer individuals in the catch, then it was analyzed completely. In cases where it was not possible to determine the total number of Antarctic krill and salps in the catch, the entire catch was weighed and, depending on the amount of krill and salps caught, a sample (from 100 mL to 1 L) was taken and extrapolated to the entire catch. A total of 3789 krill individuals were measured, and an analysis was performed for 2561 krill and 6421 salps.

### 2.6. Parasitological Studies of Antarctic Fish

Five specimens of the Antarctic deep-sea smelt *Bathylagus antarcticus* (Günther, 1878), infected with copepods, were caught in the Atlantic sector of Antarctica off the Powell basin, in the region of the northern submarine rises and on the shelf of the South Orkney Islands in February 2020. These were collected during the 79th cruise of the R/V "Akademik Mstislav Keldysh" Shirshov Institute of Oceanology of Russian Academy of Sciences (IO RAS). The fishes were caught with mid-water trawls at three stations: st. 6653 (62°26.9545′ S, 52°14.5092′ W), 2794–2882 m in depth; st. 6655 (62°28.7629′ S, 50°57.1162′ W), 730–838 m in depth; and st. 6690 (59°47.4053′ S, 50°25.5293′ W), approximately 1400 m in depth. Fish were immediately preserved in 70% ethanol. Copepods were collected from ethanol-fixed fish in the laboratory under a stereomicroscope with magnification 15–20. In the laboratory, copepods were carefully removed from the tissues of the fish, and then soaked in lactophenol for an entire day before dissection. Mesoparasitic copepods from all specimens were deposited in the Collection of Marine Parasites (CMP: 7 vouchers Nos.: 1370.Cr.40.v1; 1371.Cr.40.v2; 1372.Cr.40.v3; 1373.Cr.40.v4; 1374.Cr.40.v5; 1375.Cr.41.v1; 1376.Cr.41.v2) of the A.O. Kovalevsky Institute of Biology of the Southern Seas RAS (IBSS), Sevastopol [44].

## 3. Results

### 3.1. Trace Elements

Concentrations of all studied elements (Figures 6 and 7), except Mo, were generally low, matching the lower limit of their background concentrations in oceanic waters [16,23], or were even lower.

For some elements, the values obtained were below their detection limits: Be—<0.005, Co—<0.005, Ag—<0.01 and Tl—<0.001 $\mu g \cdot L^{-1}$. Concentrations of Se were extremely low, and slightly exceeded its detection limit (0.01 $\mu g \cdot L^{-1}$) only in a few samples from the southern and western parts of the Bransfield strait. Concentrations of V and Ni are not presented in figures, as they were distributed quite homogeneously within the studied region, and matched the background ranges for the elements reported for oceanic

waters [16,23,45]. Ranges of the values measured in the actual study were $1.17 \div 1.79$ $\mu g \cdot L^{-1}$ for V and $0.21 \div 0.39$ for Ni.

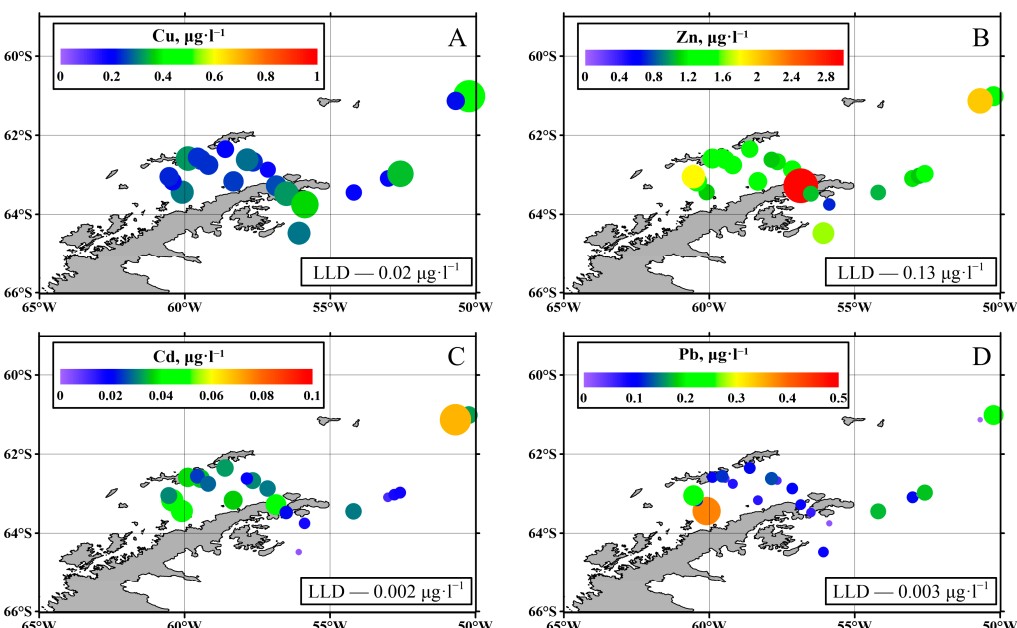

**Figure 6.** Concentrations of the main heavy metals ((**A**)—Cu; (**B**)—Zn; (**C**)—Cd; (**D**)—Pb) in surface waters of the studied area of the Antarctic region.

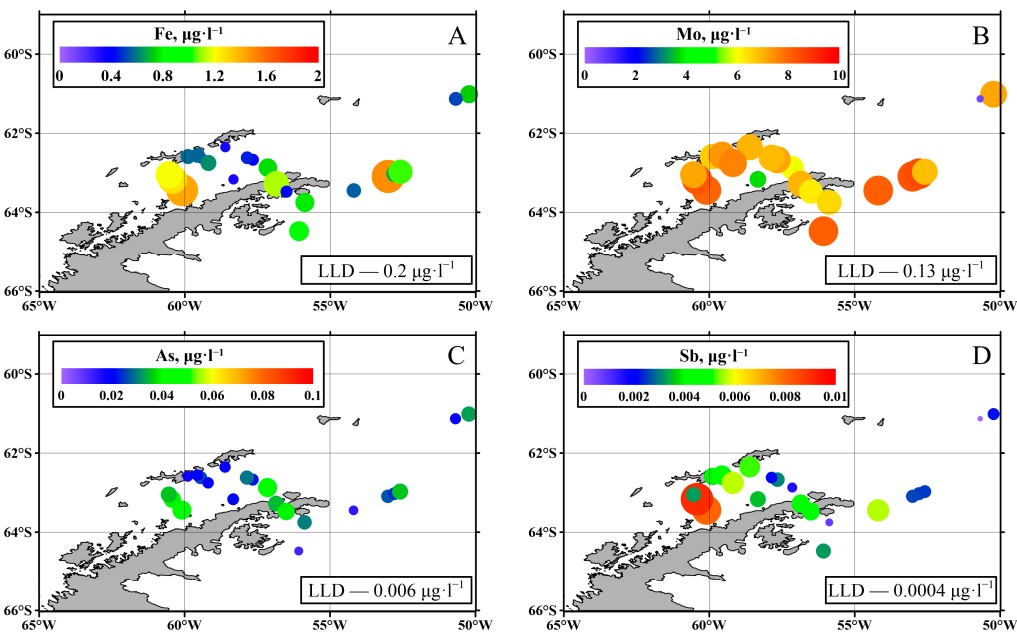

**Figure 7.** Concentrations of Fe (**A**), Mo (**B**), As (**C**), and Sb (**D**) in surface waters of the studied area of the Antarctic region.

### 3.2. Bacterioplankton–Phytoplankton Coupling and Bacterial Physiological Activity

In the Bransfield Strait, the bacterioplankton abundance varied from 0.14 to $1.03 \times 10^6$ cells $mL^{-1}$ and averaged $0.36 \pm 0.18$ (SD) $\times 10^6$ cells $mL^{-1}$. The highest concentrations of bacteria were observed in the surface layer at st. 6593, and in the 20 m layer in the northern part of the strait (Figure 8).

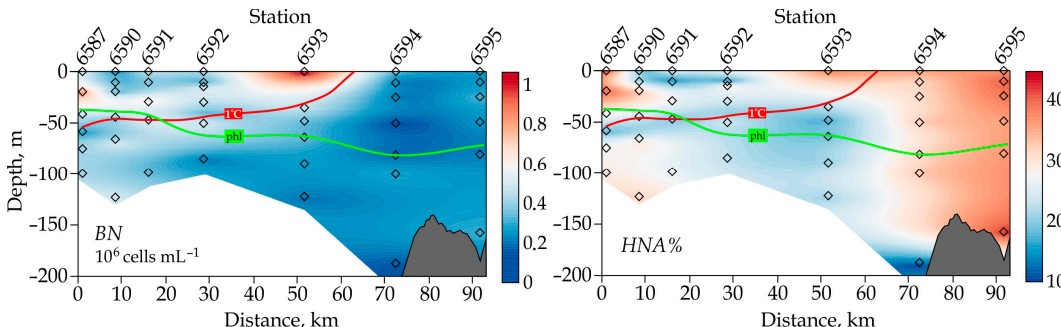

**Figure 8.** Vertical distribution of the total bacterioplankton (*BN*) and the fraction of HNA-bacteria (*HNA%)* along the transect in the Bransfield Strait during austral mid-summer, January 2020. Green and red lines represent the bottom of the photic layer (phl) and 1 °C isotherm (as a boundary between water masses), respectively.

The fraction of the HNA-bacteria was 28 ± 8% of the total abundance of bacterioplankton, and ranged from 10% to 43%. The highest HNA% values coincided with bacterioplankton peaks at st. 6587 and 6593, as well as in the southern part of the strait on the shelf of the Antarctic Peninsula (st. 6595), across the entire range of depth from the surface to the bottom (Figure 8).

The abundance of bacterioplankton exhibited a weak relationship with water temperature ($R^2$ = 0.29) and a stronger relationship ($R^2$ = 0.54) with chlorophyll fluorescence (Figure 9a,b). On the contrary, HNA% was not correlated with either temperature or chl*a* fluorescence (Figure 9c,d). Inclusion in the analysis of only physiologically active HNA-bacteria (Figure 9e,f) resulted in a significant weakening of the bacteria–phytoplankton relationship. On the contrary, (Figure 9g,h) LNA-bacteria showed a greater dependence on phytoplankton as the major source of organic matter.

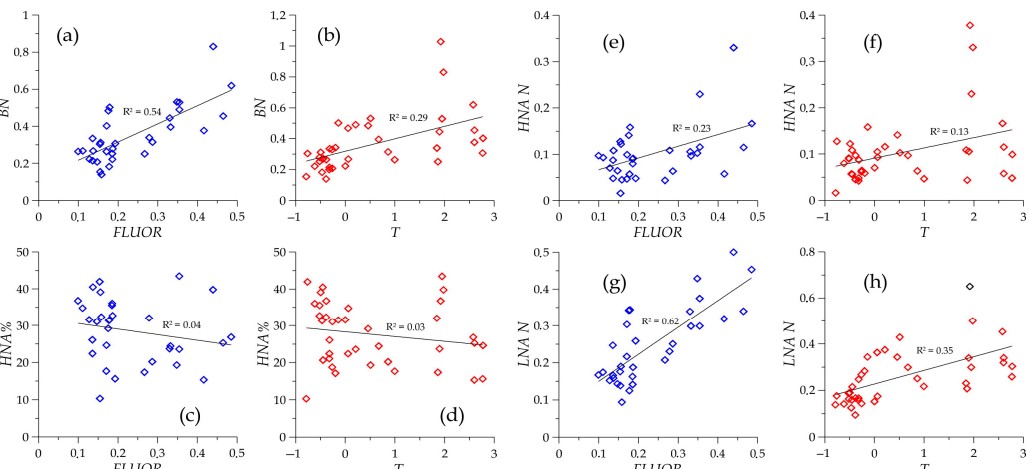

**Figure 9.** Relationships between bacterioplankton (*BN*, $10^6$ cells mL$^{-1}$), HNA-bacteria (*HNA N*, $10^6$ cells mL$^{-1}$), LNA-bacteria (*LNA N*, $10^6$ cells mL$^{-1}$), the fraction of HNA-bacteria (*HNA%*), water temperature (*T*, °C), and chlorophyll *a* fluorescence (*FLUOR*, rel. u.).

### 3.3. Spectral Bio-Optical Properties and Phytoplankton Productive Characteristics

In the austral summer 2020 (from 11 January to 4 February) in the Atlantic sector of the Antarctic, a complex investigation of bio-optical parameters, including chlorophyll *a* and phaeopigments concentration; (TChl-*a*), $a_p(\lambda)$, $a_{NAP}(\lambda)$, $a_{ph}(\lambda)$, and $a_{CDOM}(\lambda)$; and phytoplankton photosynthetic characteristics ($\phi_{max}$ and $I_k$) (Figure 10) were carried out at 37 stations in the Drake Passage, Falkland Current, Bransfield Strait, and Powell Basin.

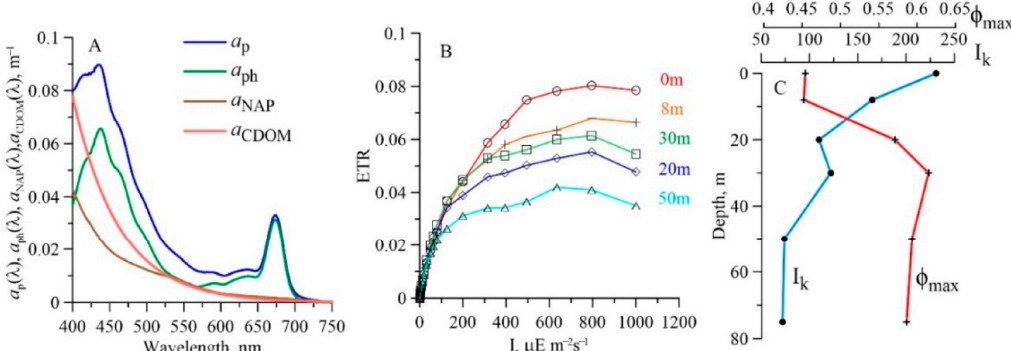

**Figure 10.** Examples of spectral light absorption coefficients of particles ($a_p(\lambda)$, m$^{-1}$), non-algal particles ($a_{NAP}(\lambda)$), phytoplankton $a_{ph}(\lambda)$, and colored dissolved organic matter($a_{CDOM}(\lambda)$) (**A**); Light curves of relative electron transport rate (ETR, rel. un.) (**B**); Vertical distribution of the saturating light intensity ($I_k$, µE m$^{-2}$ s$^{-1}$, blue line) and maximum quantum yield of photosynthesis ($\phi_{max}$, C quantum$^{-1}$, red line) (**C**).

Wide variability of the bio-optical properties (about two orders of magnitude) was observed (Figure 11). The surface TChl-*a* varied in a range from 0.20 mg m$^{-3}$ (Drake Passage) to 4.4 mg m$^{-3}$(Powell Basin). The $a_{ph}(\lambda)$, $a_{NAP}(\lambda)$, and $a_{CDOM}(\lambda)$ at the wavelength of phytoplankton physiology (maximum of light absorbance at 438 nm) varied in the surface layer from 0.0049 to 0.29 m$^{-1}$, from 0.0046 to 0.034 m$^{-1}$, and from 0.0042 to 0.15 m$^{-1}$, correspondently (Figure 11).

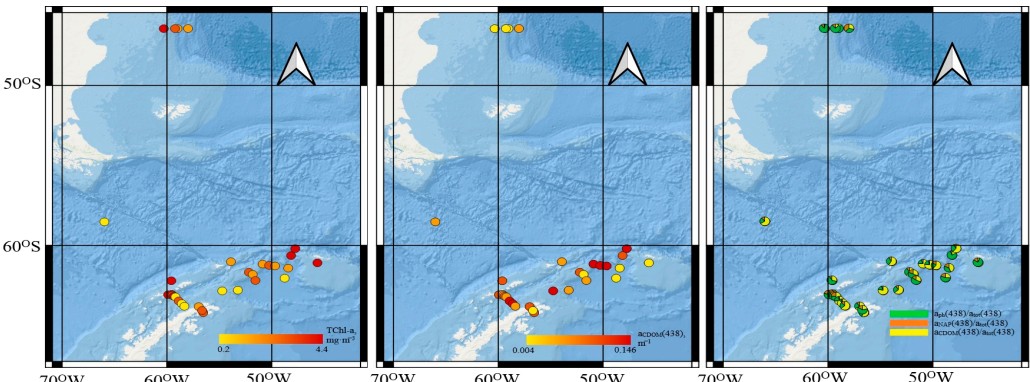

**Figure 11.** Map of the sum of chlorophyll *a* and phaeopigment concentration (TChl-*a*, left panel), light absorption coefficient of colored dissolved organic matter at 438 nm ($a_{CDOM}(438)$, middle panel), and contribution of the phytoplankton($a_{ph}(438)/a_{tot}(438)$), non-algal particles ($a_{NAP}(438)/a_{tot}(438)$), and CDOM ($a_{CDOM}(438)/a_{tot}(438)$) to total non-water absorption at 438 nm (left panel).

The $a_{ph}(\lambda)$ correlated with TChl-*a* ($a_{ph}(438) = 0.048 \times C_a^{1.2}$, $r^2 = 0.83$, $N = 126$; $a_{ph}(678) = 0.022 \times C_a^{1.2}$, $r^2 = 0.92$, $N = 126$). The power coefficients were >1. This reflects the increasing chlorophyll *a*-specific phytoplankton absorption coefficients ($a^*_{ph}(\lambda)$), with a rise of TChl-*a*. In high trophic waters (st. 6609), the $a_{ph}(\lambda)$, had a local maximum at 490 and 550 nm—the "signature" of phycobilin pigments.

As result of parameterization of non-algal particles (NAP) and colored dissolved organic matter (CDOM), absorption of S$_{NAP}$ was equal to $0.011 \pm 0.0017$ nm$^{-1}$, and that of S$_{CDOM}$ was $0.0151 \pm 0.0016$ nm$^{-1}$. The analysis showed that $a_{NAP}(\lambda)$ co-varied with both $a_{ph}(\lambda)$ and TChl-*a*, but $a_{CDOM}(\lambda)$ did not correlated with either $a_{ph}(\lambda)$ or (TChl-*a*). The contribution of phytoplankton and CDOM to the total non-water absorption at 438 nm varied significantly in the surface layer, from 7% to 88% and from 3% to 88%, respectively (Figure 11). The high phytoplankton and minimum CDOM contributions were observed in more trophic waters (TChl-*a* > 1 mg m$^{-3}$).

Analysis of the distribution of phytoplankton photosynthetic characteristics (ETR light curves, $F_{max}$, and $I_k$) revealed (Figure 8) that the $I_k$ decreased from the surface to the bottom of the euphotic zone, and that $F_{max}$ profiles showed a tendency to increase with depth.

### 3.4. Bioluminescence

Zooplankton sampling was carried out using a Bongo net [46], as well as via trawling using a Double squad net (DSN) and an Isaacs-Kidd trawl (IKMT). The Bongo net was a towed plankton net consisting of a frame with two metal rings, which had two filter cones with a mesh of 300 microns fixed onto them. The diameter of each frame ring was 60 cm [46]. DSN—a double square net with an inlet area of 1 m$^2$ and a 6 m long filter cone made of gas with a mesh size of 0.5 mm—was equipped with a water flow meter (Hydrobios, Germany) and a 24 kg wing-shaped depressor (Hydrobios, Germany). Oblique tows were carried out in layers, starting from 730 to 100 m, at vessel speeds from 2 to 3.1 knots. The towing depth was prompted by the pressure sensor readings of the Senti DT probe (Star Oddi, Iceland). An oblique tow was carried out in layers starting from 270 to 130 m, at a vessel speed of 1.7–2.3 knots. Fishing with the Bongo net showed the lowest level of effectiveness. The maximum number of specimens of *E. superba* and *S. thompsoni* were obtained using the IKMT trawl. The average size of *S. thompsoni* individuals was 30–50 mm. A large aggregation of *S. thompsoni* was registered while towing with the DSN, with an average individual size of 20–40 mm. The maximum number of salps in a catch did not exceed 400 individuals.

Aggregations of large mature krill were mainly present in oceanic areas distant from the coast, whereas larvae and juveniles were mainly concentrated in the shelf zone. The largest concentrations of larvae and juveniles of krill were noted in the Bransfield Strait on the shelf of the Antarctic Peninsula.

The maximum level of bioluminescence was registered at st.1, where it reached $272.3 \times 10^{-12}$ W·cm$^{-2}$·L$^{-1}$ at 166 m (Figure 12). The vertical structure of the bioluminescence at this station was single-maximum (Table 1), with an ocean surface temperature of 2.05 °C and salinity of 33.8 PSU.

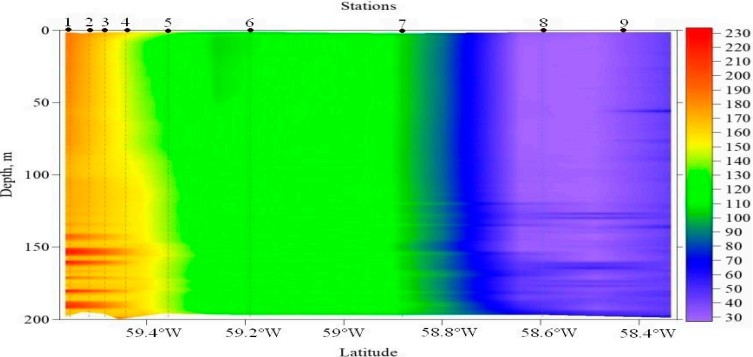

**Figure 12.** Vertical section of bioluminescence ($10^{-12}$ W·cm$^{-2}$·L$^{-1}$) in the Bransfield Strait (Figure 5).

At stations 2, 3, 4, 5, 6, and 7, the average level of bioluminescence varied from $88 \times 10^{-12}$ W·cm$^{-2}$·L$^{-1}$ to $166 \times 10^{-12}$ W·cm$^{-2}$·L$^{-1}$, the structure being evenly distributed across depth with slight peaks. At central stations of the section, a high-gradient zone was formed where the water temperature varied from 1.32 °C to 1.76 °C. The lowest level of the bioluminescent signal, with values of $18.69 \times 10^{-12}$ W·cm$^{-2}$·L$^{-1}$ and $17.42 \times 10^{-12}$ W·cm$^{-2}$·L$^{-1}$ throughout the study area, were recorded at stations 8 and 9, respectively. At the same time, the coldest ocean surface temperature layer, −0.006 °C, was also recorded at station 9 (Table 1).

In the area of the archipelago of the South Shetland Islands, where the flow of water from the southern periphery of the Antarctic Circumpolar Current enters, the accumulations of salps and a low number of krill of older age groups were recorded. The distribution of *E. superba* and *S. thompsoni* coincided with the structure of the BF in the studied region (Figure 12).

**Table 1.** Mean of bioluminescence, temperature, and salinity at several stations in the region.

| Station Coordinates | Station Number | Max. Biolum. | Max. Level (m) of Biolum. | Biolum. Means | Temp, °C | Salinity, PSU |
|---|---|---|---|---|---|---|
| 62°33.48′ S 059°33.85′ W | 1 | 272.3 | 166 | 149.57 | 2.05 | 33.8 |
| 62°34.96′ S 059°31.92′ W | 2 | 159.4 | 176 | 88.4 | 1.9 | 33.8 |
| 62°35.93′ S 059°29.42′ W | 3 | 169.87 | 179 | 165.84 | 1.76 | 33.85 |
| 62°36.89′ S 059°27.30′ W | 4 | 190.8 | 198 | 166.3 | 1.88 | 33.9 |
| 62°40.01′ S 059°22.01′ W | 5 | 115.67 | 113 | 100.72 | 1.32 | 34.11 |
| 62°44.86′ S 059°11.89′ W | 6 | 110 | 179 | 107.81 | 1.38 | 34.1 |
| 62°54.04′ S 058°53.23′ W | 7 | 109 | 111 | 101.59 | 0.68 | 34.18 |
| 63°02.14′ S 058°35.66′ W | 8 | 81.4 | 128 | 18.69 | 0.07 | 34.19 |
| 63°10.01′ S 058°20.05′ W | 9 | 55.93 | 152 | 17.42 | 0.07 | 34.19 |

### 3.5. Antarctic Krill

Our research showed that krill abundance ranged from 0 to 537 ind·1000 m$^{-3}$, and biomass ranged from 0 to 331.8 g·1000 m$^{-3}$. The maximum catch by the Isaacs-Kidd trawl was 33 kg. Significant numbers of large, immature krill were found southeast of James Ross Island, where krill numbers averaged 357 ind·1000 m$^{-3}$. The third-largest catch was in the Bransfield Strait, with a krill count of 321 ind·1000 m$^{-3}$ (Figure 13).

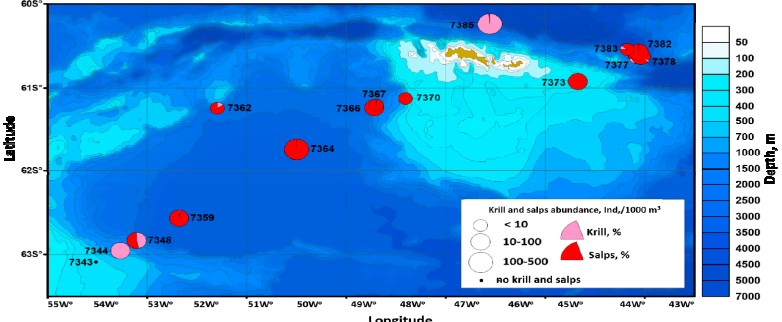

**Figure 13.** Spatial distribution of the number of accumulations (ind·1000 m$^{-3}$) of Antarctic krill and salps (%) in the Powell Basin and the area of the South Orkney Islands, according to results from the 87th cruise of R/V "Akademik Mstislav Keldysh" in 2022.

The bulk of the catch consisted of juvenile fish.

### 3.6. Parasitological Research

Five specimens of sphyriid and two specimens of pennilid copepods were identified on five exemplars of *Bathylagus antarcticus*. Only one species of copepod parasitized on infected fish. Five individuals of mature egg-bearing copepod females were found on three specimens of *B. antarcticus*, which morphologically corresponded to the *Paeonocanthus antarcticensis* (Hewitt, 1965) (Figure 14A–D). In the cephalothorax and part of the neck, copepods of this species penetrated deep into the muscles, eventually reaching the abdominal cavity and penetrating into the livers of the fish.

In the two other specimens of *B. antarcticus*, two mature, egg-bearing female copepods were found, which, according to their morphometric character, correspond to the diagnosis of the genus *Sarcotretes* Jungersen (Figure 14E–G). Among the four valid species of copepods of this genus, the species *S. scopeli* Jungersen is the closest to the studied individuals. It should be noted that the studied female copepods, *Sarcotretes* sp. from *B. antarcticus*, were gravid, with smaller body sizes than the *S. scopeli* copepods parasitizing *Bathylagus* sp. from the Southern Ocean (13.4 mm vs. 21.15 mm sensu [47]), as well as than myctophid fishes caught in southwestern Greenland, the Scotian Shelf, and off New Jersey (13.4 mm vs. 16–25 mm sensu [48]).

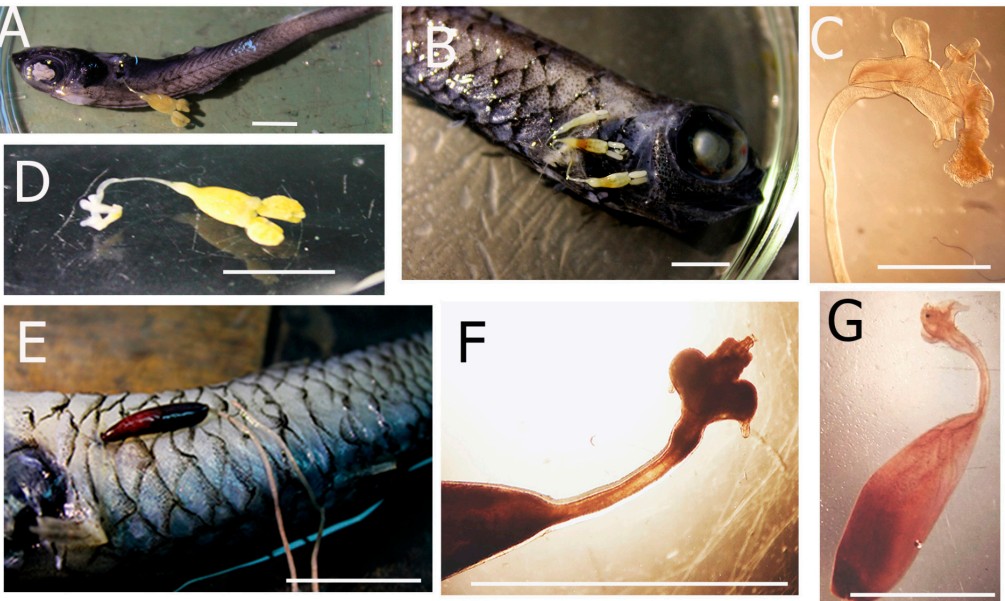

**Figure 14.** *Paeonocanthus antarcticensis* (Hewitt) (**A–D**) female and *Sarcotretes* sp. (**E–G**) female on *Bathylagus antarcticus* Günther. (**A,B**)—Attached copepods on fish body, cephalothorax, and neck, embedded in the body muscles; (**C**)—cephalothorax copepods after being soaked in lactophenol; (**D**)—live copepod with egg sac, after extraction from the fish; (**E**)—attached copepod with implanted cephalothorax in fish heart; (**F**)—live copepod cephalothorax after extraction from the fish; (**G**)—whole copepod after being soaked in lactophenol. Scalebars: (**A,B,D,E**)—1 cm, (**F, G**)—0.5 cm, (**C**)—0.2 cm.

Observed differences are likely related to the host variability of these copepods; however, further studies of a larger number of *Sarcotretes* specimens from *Bathylagus* in Antarctica are needed to confirm this statement. The cephalothorax of discovered copepods of this genus penetrated through all the organs and muscles of the fish, and directly into the cavity of the heart.

## 4. Discussion

### 4.1. Trace Elements

The measured concentrations of Zn, Cd, Pb (Figure 6), Fe, As, and Sb (Figure 7) were generally higher in samples from the western section in the Bransfield Strait, with a slight decrease in their content eastward through the Strait. This trend is highly likely to reflect the input of the enhanced content of these elements from the west of the Strait with their further dilution with Weddell Sea waters, which generally have lower concentrations of the studied metals and metalloids [22,23]. This concept is in good agreement with studies of water mass transport through the Strait [18,21,49]. The most evident trends were obtained for Fe, Pb, and Sb. Further analyses of elements' distribution within the entire studied region indicates that internal sources of Fe (Figure 7A) and Zn (Figure 6B) were likely to be present in the Weddell Sea, as enhanced values were observed for both elements in the north west part of the Sea. Perhaps this is due to the influx of elements into the sea environment during the melting of polar glaciers [16,50]. The input of considerable quantities of Fe in suspended form, sourced from the shore lands of islands and the Antarctic continent, caused leaching further toward the solution, and then assimilation with phytoplankton species was observed for the South Scotia Ridge region (NW Weddell Sea) [51]. The almost homogeneous distributions of Mo (Figure 7B) and V in the studied area were quite typical for oceanic ecosystems [52,53], since these elements reveal conservative behavior in oxygenated marine waters [45]. The average values of Mo (6.76 µg·L$^{-1}$) and V (1.52 µg L$^{-1}$) were in agreement with those reported for sea waters of the World Ocean [45,52,53]. The lack of Mo in two samples (Figure 7B) may be explained by local hypoxia due to the



commencing storms, causing the mixing of highly oxygenated surface waters with less oxygenated underlying water layers, since Mo has been shown to be sensitive to decreases in oxygen concentration [53]. The further analysis of the particulate material sampled at the same stations of the cruise may clarify this point.

A comparison of the results obtained in 2020 [22] and 2022 on the concentration of trace elements in the surface waters of the Bransfield Strait showed an increase in the concentrations of Mo and V in 2022 by 2.4 and 2.3 times, respectively. Additionally, in 2022, an excess of the average concentrations of Mo and V in the sea water of the Bransfield Strait by 1.6–1.7 times, respectively [54], was noted.

### 4.2. Bacterioplankton–Phytoplankton Coupling and Bacterial Physiological Activity

We have revealed that the highest bacterioplankton abundances can be found in the northern part of the strait, where the water masses dominate with Bellingshausen Sea influence, water temperatures of >1 °C, and salinities of <34.1 psu [55,56]. The low boundary of the water mass is well-marked by the 1 °C isotherm (Figure 9). High Chl-*a* concentrations, found at the same stations, agreed well with the earlier long-term observations: well-stratified waters off the South Shetland Islands are associated with a shallower photic layer and higher Chl-*a* concentrations at or near the surface [57]. Thus, the Bransfield Strait bacterioplankton demonstrated good coupling with phytoplankton, the major supplier of organic carbon in the Southern Ocean, under the conditions of negligible input of all ochthonous organic matter to the coastal waters [58]. According to our results, the fraction of HNA bacteria was relatively small compared with other regions of the World Ocean. This agreed well with other data [59,60] and supported the hypothesis about the important role of other mechanisms (such as bacterivory and phage infection) in controlling Antarctic bacterioplankton [61–63]. Our attempt to reveal a stronger relationship between HNA bacteria and phytoplankton was unsuccessful, indicating that in the Antarctic waters, bacterial abundance may be closely controlled by loss processes rather than resource supply [64].

### 4.3. Spectral Bio-Optical Properties and Phytoplankton Productive Characteristics

The investigated water region had high spatial heterogeneity in the distribution of the bio-optical parameters caused by dynamics and hydrological structure of the waters [65,66]. The water dynamics resulted in variability of the main environmental factors—light, temperature, and nutrients supply—controlling the abilities of phytoplankton and its photosynthetic characteristics [67,68]. The revealed links between $a_{ph}(\lambda)$ and TChl-*a* differed from their relationship in the Antarctic waters [69] and World Ocean [70] by the higher power coefficients (>1). This reflected the increasing levels of $a^*_{ph}(\lambda)$, with a rise in TChl-*a* caused by a dominance of small-celled species of phytoplankton in warmer (surface temperature >3 °C) and trophic waters (TChl-*a*> 1 mg m$^{-3}$). The "signature" of phycobilin pigments on the $a_{ph}(\lambda)$ justified an increasing of cryptophyte abundance in phytoplankton and a shift in dominating taxon to Cryptophyte [71]. The collected bio-optical data was the basis for a validation of the standard satellite products, for parameterization of absorption by all optically active components, and for refining the developed three bands algorithm [72] for Antarctic waters. The measured bio-optical properties and quantum efficiency of photosynthesis will be used for the calculation of primary production (PP) by a full spectral approach, using spectral downwelling irradiance ($E_d(\lambda)$), $a^*_{ph}(\lambda)$, and $\phi_{max}$, $I_k$) [73]. The spectral PP takes into account the effect of global warming on the species and size structure of phytoplankton, with its state caused by adaptation to ambient light and nutrient availability (via $a^*_{ph}(\lambda)$, and $\phi_{max}$, $I_k$) [36,70,74], as well as the effect of ice melting on the variability of the NAP and CDOM content (via $E_d(\lambda)$) [75].

### 4.4. Study of Bioluminescence

Thanks to the use of hydrobiological complexes and the method of multiple soundings, there was an opportunity to register the fields and the structure of accumulations of Antarctic krill, salps, and other luminous organisms in the 0–200 m layer in Antarctic

waters. It was stated that one of the main features of the vertical structure of Antarctic bioluminescence fields (BFs) is their stratification, determined both by the parameters of the pelagic community and by the characteristics of the water masses. The depth of the layer or layers of maximum luminescence intensity (MLIL) and their number are also important characteristics of the BF. During the study period, the MLILs recorded at almost all stations were below the isobath of 100 m. Intense outbreaks of bioluminescence in the area of the South Shetland Islands archipelago (with a low abundance of krill) can be associated with an increase in the abundance of salps, which are capable of generating outbreaks of such potential. A rapid increase in the number of salps was likely due to their rapid asexual reproduction in ice-free spring waters [2]. The increase in the intensity of the bioluminescence field in the studied waters was likely due to an improvement in the supply of nutrients to the waters and a significant increase in the content of digestible organic matter. Moving to the south, the intensity of bioluminescence was noticeably decreasing, which was likely due to the horizontal and vertical structure of the waters. A significant decrease in temperature and an increase in salinity have a substantial impact on the distribution of luminous organisms [76]. Low temperatures seem to have a negative effect on the development of salps; therefore, they are not found in the southern regions. The largest concentration of larvae and juveniles of krill, which, unlike accumulations of salps, cannot produce a powerful bioluminescent signal, was registered on the shelf of the Antarctic Peninsula [76].

### 4.5. Antarctic Krill and Salpa Thompsoni

Against the background of a decline in Antarctic krill abundance in recent years, there has been an intensive development of *Salpa thompsoni*, the main food competitor of Antarctic krill. Catastrophic outbreaks of this species due to climate change, observed since 1975 and up to the present time, raise reasonable concerns about possible adverse changes in the pelagic ecosystem of the Southern Ocean [77–83]. These jelly planktonic organisms change the structure of trophic chains, which affects the ecology of many animals, including fish, birds, and mammals. According to our experimental determinations of energetic metabolism [84], the dietary requirements of *S. thompsoni* are up to two orders of magnitude (depending on the size of tunicates) greater than those of Antarctic krill [85]. The mechanism of salps penetration to high latitudes is based on the meridional gyres exchange through the Antarctic Circumpolar Current front [86]. Under the conditions of large-scale atmospheric processes, this mechanism is universal for all sectors of Antarctica [2]. It appears to be key to the awareness of the expected serious changes in the Antarctic ecosystem under the influence of global climate change, displacement of polar fronts, and main continental margin currents, and it is most pronounced in the Atlantic sector [87]. During the period of the IBSS surveys in the Atlantic sector, the R/V "Ernst Krenkel" (1998) and Horizon (2002) recorded potentially commercial-scale accumulations of krill, with stocks estimated at millions of tons [85]. In the time of the 87th expedition of RV "Akademik Mstislav Keldysh" in 2022, it was discovered that the krill fields had been replaced by accumulations of salps. Their numbers ranged from 0 to 202 ind·1000 m$^{-3}$ and from 0 to 72 g·1000 m$^{-3}$, respectively. It is thought that the possible catastrophic consequences for the Antarctic ecosystem will involve three main factors [2]—mechanical removal from the base of krill habitat; suppression of its population development due to inhibition of reproduction in the crustacean, whose spawn develops on the polluted bottom of shallow waters; and through food competition with salps.

### 4.6. Parasitological Research

Parasitic copepods are common fish parasites, and can have pathogenic effects on the host, sometimes causing their death. This can result in economic problems in the fishing industry [88–91]. Currently, there is limited information about ecto- and mesoparasitic crustaceans from fish of the Southern Ocean [47,92,93]. According to [93,94], only nine species of copepods, from seven genera and four families, have been identified from

fish from this region between 1966 to 2015. In the Southern Ocean, only two species of sphyriid and pennilid copepods have previously been found in the bathypelagic fish *Bathylagus* Günther. In *B. antarcticus,* in the Pacific Sector of Antarctica, D'Urville Sea (65° S, 139°59.6′ E), the copepod *Paeonocanthus antarcticensis* from the family Sphyriidae Wilson was observed [92]. In the Antarctic, *Sarcotretes* copepods were mainly recorded in the gills of mesopelagic fishes (Myctophidae) [93], and the only record of *S. scopeli* was recorded in a bathypelagic fish, *Bathylagus* sp., in the Indian Ocean Sector of Antarctica, Davis Sea, and Lützow-Holm Bay [47,48]. Before our studies, there had been no repeated findings of copepods of the *Paeonocanthus* and *Sarcotretes* genera in bathypelagic fishes in Antarctic waters. The discovery of copepods of these two genera in endemic bathypelagic fish of the Southern Ocean in a new area, the Atlantic sector of Antarctica off the Powell basin, significantly expands our know ledge about their range and the species composition of their fish hosts. According to [90], representatives of the genus *Sarcotretes* can penetrate into the liver when penetrating the cephalothorax and part of the neck into the body cavity of the fish, which leads to a decrease in their volume in comparison with uninfected fish. In addition, V.N. Kazachenko [90] noted the penetration of cephalothorax copepods of this genus into the abdominal cavity near the intestines of fish, which led to a decrease in the intestinal lumen and, as a result, a disruption in the functioning of this organ. During our study, we found a more pathogenic influence of copepods of the genus *Sarcotretes*, namely the penetration of the cephalothorax and part of the neck by *Sarcotretes* sp., into the heart cavity of *Bathylagus antarcticus,* leading to the parasite feeding on the blood of the fish. We also noted a deep penetration of the cephalothorax of another copepod species, *Paeonocanthus antarcticensis*, into the liver of *B. antarcticus*. In this casem the muscles at the site of penetration were very loose, and the liver was slightly deformed. The formation of tumors at the sites of penetration of copepods of both species was not found in *Bathylagus*.

## 5. Conclusions

In general, the study of trace element distribution in surface waters of the Atlantic Sector of Antarctica in 2022 ensured that the studied water area was the reference ecological background region in terms of content of trace elements, including heavy metals. However, comparison of the results obtained in 2020 [22] and 2022 on the concentration of trace elements in the surface water of the Bransfield Strait showed an increase in the concentrations of Mo and V in 2022 by 2.4 and 2.3 times, respectively. Such an increase did not result in excess of any maximum permissible concentrations, but led to the average concentrations of Mo and V in the sea water exceeding the recommended "target values" (considered absolutely safe for a marine environment) of these elements by 1.6–1.7 times [49]. This testifies to the need to continue monitoring chemoecological studies in this region of the Southern Ocean, in order to identify the sources of trace elements entering the studied area.

In Antarctica, the primary production of phytoplankton determines not only the matter and energy flow up to the higher trophic levels, including Antarctic krill, but also the dependence of heterotrophic bacteria on algae-produced labile dissolved organic matter. In 2020 the highest abundance of bacterioplankton (up to $1.03 \times 106$ cells mL$^{-1}$) was observed in the northern part of the Bransfield Strait in the warmer and less salty water mass under Bellingshausen Sea influence, just where the phytoplankton bloom was registered. A lower fraction of physiologically active HNA bacteria ($28 \pm 8\%$ on average) and their weaker coupling with phytoplankton indicated that the Bransfield Strait bacterioplankton seemed to be closely controlled by predatory and viral mortality rather than organic carbon supply by phytoplankton. The variability in the bio-optical properties of the Southern Ocean was assessed in the austral summer of 2020. The high spatial heterogeneity in the distribution of light absorption by all optically active components was also revealed. The collected bio-optical data present the scientific basis for the development of regional bio-optical algorithms for operative assessment of the current state of the Southern Ocean's pelagic ecosystem, as well as forecasting changes due to global warming. Satellite data provide primary production estimates with high spatial and temporal resolution. The

combination of in situ data and data calculated using satellite algorithms on the abundance of phytoplankton (the concentration of chlorophyll *a* is used as a biomass marker) and its production activity will make it possible to further study the patterns of formation of primary production and food supply of krill in contemporary conditions, as well as the basis of established relationships, to predict possible changes due to global warming.

The findings of mesoparasitic copepods of two genera (*Paeonocanthus* and *Sarcotretes*) in bathypelagic fish endemic to the Southern Ocean in a new area, the Atlantic sector of Antarctica off the Powell basin, significantly expands our knowledge about their geographical distribution and the species composition of their definitional fish hosts—*Bathylagus spp*. The conducted research and the obtained results are aimed at supporting management decisions regarding the exploitation and conservation of the unique ecological and biological resources of Antarctica. This research is also the basis for further research by the IBSS in the Atlantic sector of the Antarctic.

**Author Contributions:** Conceptualization, N.M. (Natalia Mirzoeva), E.S. (Ernest Samyshev), T.C., V.M. (Vladimir Mukhanov), T.P. and A.M.; methodology and validation, N.M. (Natalia Minkina), V.P., E.S. (Evgeny Sakhon), E.S. (Elena Skorokhod), N.M. (Natalia Moiseeva), E.C., O.C.-P., A.P. and T.E.; investigation, T.P., V.M. (Vladimir Mukhanov), N.M. (Natalia Moiseeva), V.P., O.C.-P., E.S. (Elena Skorokhod), A.P., E.C., E.S. (Evgeny Sakhon) and A.M.; formal analysis and writing—original draft preparation, N.M. (Natalia Mirzoeva), E.S. (Ernest Samyshev), T.C., V.M. (Vladimir Mukhanov), T.P., A.M., V.M. (Victor Melnikov), V.P., N.M. (Natalia Minkina), N.M. (Natalia Moiseeva), A.P. and L.M.; writing—review and editing, N.M. (Natalia Mirzoeva), T.P., E.S. (Ernest Samyshev), T.C., V.M. (Vladimir Mukhanov), V.P. and A.M. All authors have read and agreed to the published version of the manuscript.

**Funding:** This study was conducted within the framework of the Russian state task No. 121090800137-6 "Comprehensive studies of the current state of the ecosystem of the Atlantic sector of Antarctica".

**Informed Consent Statement:** Not applicable.

**Data Availability Statement:** All data used in this study are available upon request from the corresponding author.

**Acknowledgments:** The authors are sincerely grateful to the crew of the R/V "Academician Mstislav Keldysh" for their help with the fieldwork. Special thanks to the head of the expedition, Morozov E. G., and the deputy head of the expedition, Molodtsova T. N. (IO RAS) for supporting and organizing trawling operations. The authors thank Bitiutskii D. G., Leading Researcher, and Usachev S. I., Chief specialist, (AZNIIRH) for their assistance in sampling processing for all participants of the sea expedition. The authors are sincerely grateful to the director of the FRC IBSS, Gorbunov R. V. for the organization of these scientific researches.

**Conflicts of Interest:** The authors declare no conflict of interest.

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
