# Peer review of "Current Assessment of Water Quality and Biota Characteristics of the Pelagic Ecosystem of the Atlantic Sector of Antarctica: The Multidisciplinary Studies by the Institute of Biology of the Southern Seas"

_water, doi:10.3390/w14244103_

Round 1
Reviewer 1 Report (New Reviewer)
This paper could be greatly strengthened by a thorough review of English and grammar. I found sections difficult to read and understand and many acronyms not defined.
I also feel the parasite section would be better placed as a separate paper or short note
I have made extensive comments on the attached file for consideration by all authors

Author Response
The authors of the work express their deep gratitude to the reviewer for the great work in the careful study of the text of the manuscript. Based on the comments of the reviewer, which the authors accepted almost in full, the grammar of the English language was improved, the semantic presentation of the text was optimized.
After revision, the authors gave the Manuscript to be proofread by a native English speaker (a colleague from Canada), who was satisfied with the quality of the grammatical presentation of the material.
- The authors have optimized the title of the manuscript.
- Added and improved Figures.
- Added necessary references.
You can see all the edits made in the sent file, where the edits made are presented in the text.
Specific responses to the reviewer's comments for each area of scientific research are presented in file below.

Reviewer 2 Report (New Reviewer)
Several comments that need to be elaborated and explained in more detail by the authors as follows:
1. It is suggested that the last part of the paper title i.e.: "the complex study by Institute of Biology of the Southern Seas" is better to be deleted.
2. It should be better if the informations of water quality and biota characteristics in the study region from the previous study are explained in the intro chapter.
3. It is important to show all the sampling station maps in the method chapter e.q. sampling stations for Temperature, conductivity, Chla, ap, aph, anop, acdom, and krill (not only trace elements and bacterioplankton sampling stations).
4. It is need to be consistent to present the coordinate number or symbol in all map i.e., either to write latitude in 60 S or -60 (choose one).
5. In Figure 4, it will be better if the authors provide information for the vaue of the low limit of the trace elements.
6. In the discussion chapter and conclusion, it is needed to explain in more detail how the autors concluded that the krill productivity depends on primary productivity (chla concentration). Yet, the sampling stations of krill and chla concentratio were totally not match one to another.
Author Response
The authors of the work express their deep gratitude to the reviewer for the great work in the careful study of the text of the manuscript. Based on the comments of the reviewer, which the authors accepted almost in full, the grammar of the English language was improved, the semantic presentation of the text was optimized.
After revision, the authors gave the Manuscript to be proofread by a native English speaker (a colleague from Canada), who was satisfied with the quality of the grammatical presentation of the material.
- The authors have optimized the title of the manuscript.
- Added and improved Figures.
- Added necessary references.
You can see all the edits made in the sent file, where the edits made are presented in the text.
Specific responses to the reviewer's comments for each area of scientific research are presented below.
The authors of the work express their deep gratitude to the reviewer for the great work in the careful study of the text of the manuscript. Based on the comments of the reviewer, which the authors accepted almost in full, the grammar of the English language was improved, the semantic presentation of the text was optimized.
After revision, the authors gave the Manuscript to be proofread by a native English speaker (a colleague from Canada), who was satisfied with the quality of the grammatical presentation of the material.
- The authors have optimized the title of the manuscript.
- Added and improved Figures.
- Added necessary references.
You can see all the edits made in the sent file, where the edits made are presented in the text.
Specific responses to the reviewer's comments for each area of scientific research are presented below.
The authors of the work express their deep gratitude to the reviewer for the great work in the careful study of the text of the manuscript. Based on the comments of the reviewer, which the authors accepted almost in full, the grammar of the English language was improved, the semantic presentation of the text was optimized.
After revision, the authors gave the Manuscript to be proofread by a native English speaker (a colleague from Canada), who was satisfied with the quality of the grammatical presentation of the material.
- The authors have optimized the title of the manuscript.
- Added and improved Figures.
- Added necessary references.
You can see all the edits made in the sent file, where the edits made are presented in the text.
Specific responses to the reviewer's comments for each area of scientific research are presented below.
The authors of the work express their deep gratitude to the reviewer for the great work in the careful study of the text of the manuscript. Based on the comments of the reviewer, which the authors accepted almost in full, the grammar of the English language was improved, the semantic presentation of the text was optimized.
After revision, the authors gave the Manuscript to be proofread by a native English speaker (a colleague from Canada), who was satisfied with the quality of the grammatical presentation of the material.
- The authors have optimized the title of the manuscript.
- Added and improved Figures.
- Added necessary references.
You can see all the edits made in the sent file, where the edits made are presented in the text.
Specific responses to the reviewer's comments for each area of scientific research are presented below.
The authors of the work express their deep gratitude to the reviewer for the great work in the careful study of the text of the manuscript. Based on the comments of the reviewer, which the authors accepted almost in full, the grammar of the English language was improved, the semantic presentation of the text was optimized.
After revision, the authors gave the Manuscript to be proofread by a native English speaker (a colleague from Canada), who was satisfied with the quality of the grammatical presentation of the material.
- The authors have optimized the title of the manuscript.
- Added and improved Figures.
- Added necessary references.
You can see all the edits made in the sent file, where the edits made are presented in the text.
Specific responses to the reviewer's comments for each area of scientific research are presented in file below.

Round 2
Reviewer 1 Report (New Reviewer)
Thank you for considering the comments. Your paper is much improved.
This manuscript is a resubmission of an earlier submission. The following is a list of the peer review reports and author responses from that submission.
Round 1
Reviewer 1 Report
This study provides insights into the status of Antarctic pelagic ecosystems in the Atlantic sector of the Southern Ocean based on two recent Antarctic cruises with the participation of members of the Russian Academy of Sciences. Data covered trace elements, chl-a, bacterioplankton, abundances of krill and salps up to fish parasites. The article has a wide range of data and it can be of interest for the scientific community, for example in comparison with other sectors of the Southern Ocean or with data collected by other Antarctic programs. However, it is often difficult to understand the links among the single results as these encompass many different disciplines (physical-biological oceanography, microbiology, zoology and ecology). The ms requires major revisions before acceptance for publication on the journal
1) The authors should better underline the significance of their findings in relation with previous/other data to represent the overall status of Antarctic pelagic marine ecosystems, as their single results seem often not too linked among each other.
2) As many data are presented in the article, the authors could provide the data sets as supplementary material to make them available to the scientific community. Es. depths considered for water samples (l. 103)
3) In the Methods, a paragraph dedicated to statistical analysis of the data is missing.
4) The conclusions should be rewritten are some paragraphs are identical to the Discussion section: use the Conclusion section to sum up the relevant points of the overall findings obtained to represent the Antarctic marine ecosystems
5) A careful revision of the text is recommended, as many typos, missing space and errors are present. Some other minor revisions are reported below:
l. 14: change to: two sea expeditions
l. 36: Change to: differentiated
l. 35-36: Give examples of these different regions in Antarctic waters
l. 52: In the aim, add that the assessment is referred to the outcomes obtained by the Russian Academy of sciences through two recent cruises, as at this stage it could be seen as a review of the literature instead.
l. 58, l.62 etc: change to: analysis
l. 63: change to: salp
l. 64: change to: (Foxton, 1961), add also for Antarctic krill. Not necessary to repeat further in the text
l. 84: space is missing
l. 86: change to: were stored
l. 90: change to: is described
l. 103: bottles?
l. 122: is the method “Salpa”described here new? In this case, the authors should stress this in the abstract to underline the novelty of the technique
l. 135: change to: bioluminescence
l. 137: change to: various
l. 171: add parentheses for the name and year
l. 257: change to E. superba and S. thompsoni
l .262-264: add citations here
l. 264-268: description of the results could be more informative, add numbers as done in paragraph 3.4
l. 269: “completely coincide” does not give the idea of the scientific value, could you add statistical analysis such as correlation relationship?
l. 347-363: move to methods/acknowledgements
l. 363-365: sum up the previous sentences in a short sentence here to introduce the Discussion section.
l. 374: change to warmer
l. 383: it seems speculative, please rephrase
l. 385: what do you mean by “in the particulate form”?
l. 391: oceanic waters from which region?
l. 493-494: add references
l. 508: what do you mean by potentially? Change to accumulation
l. 516: add references
Check throughout the main text if acronyms have been well introduced as full term in the first entry (e.g. HNA bacteria, RAS, ICP-MS, DSN) and remove full terms later in the text as only acronyms can be used later.
In Figure 9, the images should be referred to A and B.
When chlorophyll data are mentioned in the text, please refer always to the type of chlorophyll (e.g. chl “a”)
Reviewer 2 Report
The ms. submitted is the result of several expeditions done in Antarctic waters by the R/V “Akademik Mstislav Keldysh by different researchers in each campaign. It includes plankton bioluminescence data, trace element analysis sampled during the leg 87 (year 2022) as well as oceanographic data (CTD profiles, bacterioplankton abundance and analysis of fish parasites) during the leg 79 (2020). There are some important and valuable information collected during these cruises. However, the main problem is the absolute lack of coherence throughout the entire ms. There is a lack of structure, there are no clear objectives nor hypothesis. The ms. is basically a collection of data obtained by different researchers and put together without a consistent organization. There is no connection between the bioluminescence of plankton, macroplankton and bacteria abundance and fish parasites (collected on different field campaigns and different stations). Therefore, the manuscript is difficult to follow also needs to be revised thoroughly by a native English speaker for correctness and style.
The title “Comprehensive studies of the current state of the ecosystem of the Atlantic sector of Antarctica” needs to be modified. It does not represent the scope of this study. It does not represent the current state of the ecosystem and is rather a snapshot of different studies developed mainly in the western Antarctic Peninsula and some stations close to the South Orkney Islands.
In the current form the ms. does not reach the standards to be published. I encourage the authors to select those results that might have a temporal-spatial and ecological coherence to rewrite a comprehensive ms. This could be: 1.- Concentration and distribution of trace elements and its relationship with water masses along the Bransfield strait (this could also include the data of phytoplankton fluorescence). 2. Another paper could deal with the validation of the use of bioluminescence systems to study zooplankton abundance. 3. While other potential paper (i.e. short note) could deal with fish parasites.